# Psychometric Properties of the PLAY*self* in a Cohort of Secondary School Student-Athletes

**DOI:** 10.3390/ijerph21101294

**Published:** 2024-09-28

**Authors:** Monica R. Lininger, Hayley J. Root

**Affiliations:** Department of Physical Therapy and Athletic Training, Northern Arizona University, Flagstaff, AZ 86011, USA; hayley.root@nau.edu

**Keywords:** physical literacy, physical activity, sports, youth

## Abstract

Background: Physical literacy is the motivation, confidence, physical competence, knowledge, and understanding, enabling individuals to value and take responsibility for engagement in physical activities for life. While tools exist to measure physical literacy in most populations, the psychometric properties of the Physical Literacy Assessment for Youth (PLAY) tool in an older adolescent age group are currently unknown. The purpose of this work was to determine the psychometric properties of the PLAY tool, specifically the PLAY*self,* in an older adolescent age group (~14–18 years). Methods: One hundred and fifty-one secondary school in-season student-athletes completed the PLAY*self*, with construct validity assessed using an Exploratory Factor Analysis (EFA). Results: Results from the EFA yielded a 7-factor model across the three subsections (environment, physical literacy self-description, relative rankings of literacies) of the PLAY*self*, all with acceptable levels of internal consistency. Conclusions: The PLAY*self* produced acceptable estimates for construct validity and reliability, making it a useful tool for measuring physical literacy in secondary school student-athletes.

## 1. Introduction

Physical inactivity is a growing public health concern, particularly in youth. Between 2017 and 2020, the prevalence of obesity in the United States was over 20% in children and adolescents aged 6–19 [1]. This elevated prevalence is due to youth getting less than the recommended 60 min per day of physical activity and engaging in more than the suggested amount of screen time daily [2]. A possible explanation for the decreased physical activity rates could be that children do not have the necessary physical literacy to be physically active [3,4]. Physical literacy is correlated with sport and exercise participation and well-being metrics [5,6]. Physical activity and physical literacy levels in children and adolescents are strongly related to behaviors in adulthood [7]. Higher levels of physical activity engagement in youth can have positive long-term health benefits such as better bone health, improved cardiovascular health, and continued physical activity into adulthood [8]. More recently, Houser et al. found that children with higher levels of physical literacy before coronavirus disease (COVID-19) were better protected against the decline of physical activity during the pandemic [9].

Physical literacy is the “motivation, confidence, physical competence, knowledge and understanding to value and take responsibility for engagement in physical activities for life” [10,11,12]. There has been considerable debate over the definition of this emerging construct [10,13,14], but currently, the most agreed-upon domains include the following: physical (the ability to move and quality of movement), cognitive (knowledge and understanding the importance of physical activity for life), affective (motivation and confidence to engage in physical activity), and behavioral (actual engagement in physical activity) [14,15]. Each domain significantly influences the others. Lower levels of physical literacy are associated with lower cardiorespiratory fitness [3], higher percentages of fat mass [4], and lower levels of physical activity engagement [16]. The Physical Literacy Assessment for Youth (PLAY) tool suite was released in 2018 and includes various tools to assess physical literacy in children seven years and older [17]. One of the tools, PLAY*self*, was created to evaluate a child’s perception of their physical literacy. Psychometric analyses have been conducted for the PLAY*self* in younger adolescents (8–13 years of age) [18], middle school children [19], Canadian children 8–14 years of age [20], and college-age adults (18–25 years of age) [21], but not with adolescent athletes at the secondary school level (i.e., high school, approximately ages 14–18 years). With continued growth in understanding and evaluation of physical literacy [22], measurement accuracy is critical [18].

The long-term athlete development (LTAD) model is a framework to guide sport and physical activity participation, training, competition, and recovery pathways [23,24]. Developing physical literacy is one of the primary objectives of the first few stages of the LTAD model. In the later stages of the LTAD model, the “Train to Train” stage aims to build an aerobic base, speed, and strength in parallel with pubertal maturation. In this stage, physical literacy is critical, and children who lack the fundamental movement ability or skills as their peers may be discouraged from continuing participation. Athletes aged 14–18 likely fall into this stage given typical maturation and development trajectories. Therefore, it is critical to evaluate PLAY*self* in this missing population. Due to the lack of empirical evidence for adolescent athletes, it is necessary to better understand current levels of physical literacy to positively impact future health outcomes.

Adolescent athletes are in a stage of physical and emotional development, making this a critical time to assess physical literacy. Assessing physical literacy in this age group demands age-appropriate measurements. Therefore, the purpose of this study was to evaluate the psychometric properties of the PLAY*self* in a cohort of secondary school-aged athletes. It was hypothesized that the PLAY*self* would be multidimensional [8] in nature.

## 2. Materials and Methods

### 2.1. Study Participants and Study Design

We conducted a large cross-sectional study using survey methodologies to assess physical literacy, sports specialization, and social determinants of health in secondary school student-athletes recruited from three schools in the Southwestern United States. This manuscript focuses on the validation process of the larger study. Two schools from the same district had student bodies with 53–68% minority enrollment, 25% qualifying for free and reduced lunch, and 49% female students [25]. The third school, located in a different city and district, had just over 36% minority enrollment, 5% of students qualifying for free and reduced lunch, and 49% females in the student body makeup [25]. To participate, student-athletes had to be currently active in a high school sports season and between the ages of 13–18. Seven days before data collection, a packet containing the questionnaire, a parental informed consent form, and a minor assessment form were sent home to prospective participants for review by a parent or guardian. Willing parents signed the informed consent, and student-athletes aged 17 and younger signed the minor assent form. Student-athletes who were 18 could sign the informed consent form as legal adults. The school district’s research office mandated the process of sending materials home. Student-athletes were only able to participate in the study with a completed parental informed consent. All ethical practices and human protection procedures were approved by the research team’s academic home university (project # 1990341-2) and the secondary school’s district research office. 

### 2.2. Instrumentation

Demographic items included biological sex, gender, race, ethnicity, and completed grade level. The PLAY*self* [17] included four subsections: (1). environment; (2) physical literacy self-description; (3). relative rankings of literacies; and (4). fitness. For environment, the student-athlete rated their performance in six different environments (such as in a gym, on and in water, on snow and ice, and outdoors) using a 5-point Likert scale ranging from “never tried” to “excellent.” For scoring, “never tried” = 0, “not so good” = 25, “ok” = 50, and “excellent” = 100, with a maximum possible score of 600. The second subsection, physical literacy self-description, assessed 12 items using a 4-point Likert scale (“not true at all” to “very true”) to evaluate self-efficacy related to physical activity. The relative rankings of literacies allowed the student-athlete to rate their perceived importance of literary, numeracy, and physical literacy across nine items on a 4-point Likert scale with response options of “strongly disagree” to “strongly agree.” For both the physical literacy self-description and relative ranking of literacies subscales, scores were based on “not true at all” or “strongly disagree” = 0, “not usually true” or “disagree” = 33, “true” or “agree” = 67, and “very true” or “strongly agree” = 100. The maximum possible score for the physical literacy self-description subsection was 1200, and the maximum score was 900 for the relative rankings of the literacies subsection. The final subsection, fitness, was a single item (“My fitness is good enough to let me do all the activities I choose”) with options of “disagree” and “agree.” The overall PLAY*self* score was derived from the sum of each subsection, divided by the total number of items (*n* = 27), with a maximum score of 100, with higher values indicating a better self-perceived level of physical literacy [18,20,21]. The single fitness item was not included in the total scoring, which aligns with the previously validated scoring procedure.

### 2.3. Data Collection

Data collection procedures occurred between the Spring of 2023 and the Fall of 2023 to encompass two seasons for the student-athletes. Research team members met with coaches and student-athletes to describe the study and invite voluntary participation. In a single session, participants were asked to independently complete a hard copy questionnaire that included demographic items and the PLAY*self* assessment. A student-athlete could not complete the questionnaire more than once. In other words, a student-athlete who completed the questionnaire in the Spring could not complete it again in the Fall. Completing the entire questionnaire took no more than 15 min, and student-athletes received a $20 gift card after completing the study.

### 2.4. Data Analysis

The item “I don’t really need to practice my skills; I’m naturally good”, from the PL self-description subscale, was recoded to align with all other positively stated items. Following the recoding, each subsection was scored with totals of up to 600 for the environment subsection, 1200 for self-description, and 900 for relative rankings of literacy [21,26]. Scores from each subsection were summed and then divided by the total number of items (27) for a total physical literacy score (maximum of 100) [20]. Ceiling effects were assessed and considered present if over 80% of the sample endorsed the highest response option [21]. Calculated descriptive statistics included continuous variables with means and standard deviations and frequencies with percentages for categorical variables. 

An Exploratory Factor Analysis (EFA) was conducted to assess the factor commonalities for each of the three subsections (environment, relative ranking of literacies, and self-description). The EFA allows for the determination of the underlying dimensionality and factor structure [27], along with an estimate for construct validity [28]. Prior to conducting the EFA, polychoric correlations were calculated between all individual items [29], along with assumption testing for sphericity, sampling adequacy, and normality [30]. Overall, the sample size was adequate, with 5–10 (*n* = 151) [31] as the Self-description subsection has the highest number of items at 12. The number of factors retained was based on the following established criteria: a scree plot, eigenvalues (>1.0), and the percentage of explained variance [28,30]. If normality was violated, an EFA using principal axis factoring estimation [31] was used; if normality was not violated, an EFA with maximum likelihood estimation was utilized [32]. Oblique rotations, specifically Promax, were used in the analysis due to the possible correlation of factors [30]. The individual items were only maintained if loadings were >0.40 within each factor, and at least two items must be retained to justify a factor [28]. After the final factors were determined, internal consistency measures were calculated. Because of the ordinal nature of the data, McDonald’s Omega alpha (α) [33,34,35] point estimates were calculated for factor level reliability. Minimally, the threshold of Nunelly [36] for reliability estimates of 0.70 was used, but the authors acknowledge the weakness in creating cutoff values for reliability estimates, as noted by many experts in the field [33,34,36]. All statistical analyses were completed in the Statistical Package for the Social Sciences (IBM SPSS, Inc., 29.0, Chicago, IL, USA) with an alpha set to 0.05. 

## 3. Results

One hundred and fifty-one secondary student-athletes completed the questionnaire. There was a near-even split between males and females and equal representation across all completed grades in the sample (Table 1). Most participants self-identified as White and Not of Hispanic origin and were active in both the summer and winter seasons (126/191 = 83.4%). Student-athletes competed in a variety of secondary school sports, with most in football or basketball. This cohort perceived their abilities to be the highest for outdoor activities, with the lowest confidence rating for those on the ice (Table 2). Ceiling effects were present for “I think being active is important for my health and well-being” and “I think being active makes me happier”, with over 80% of the sample selecting “very true” for these items (Table 3). Reading and math were most important to the student-athletes in school, but movement was equally as meaningful with their friends (Table 4). Overall, this is a very confident cohort, with 147 (97%) endorsing their “fitness is good enough to let me do all of the activities I choose”.

### 3.1. Physical Literacy Environment Subsection

Normality was met, as data were within two standard deviations from the mean [37]. Therefore, maximum likelihood estimation was utilized. The Kaiser-Meyer-Olkin measure of sampling adequacy was met (*p* = 0.77), and the assumptions of sphericity (Bartlett’s test χ^2^_15_ = 295.21, *p* < 0.001) and lack of an identity matrix (r = 0.11). Two distinct factors were extracted from the six individual items (Table 5), confirmed by eigenvalues higher than 1.0 and the scree plot. The first factor, *Water Environment*, included items related to activities with water and explained over 40% of the overall variance, with a high level of reliability (McDonald’s Omega α = 0.83) and inter-item correlations ranging from 0.49–0.75. The second factor, *Dry Land Environment*, accounted for an additional 13% of the variance, but the reliability estimate was lower than that of factor 1 (McDonald’s Omega α = 0.71). These items focused on activities in the gym, outdoors, or on a playground and had inter-item correlations from 0.29 to 0.55.

### 3.2. Physical Literacy Self-Description

For the second subsection, self-description, normality was also met; therefore, maximum likelihood estimation was again utilized. The Kaiser-Meyer-Olkin measure of sampling adequacy was acceptable (*p* = 0.81), along with the assumptions of sphericity (Bartlett’s test χ^2^_66_ = 333.21, *p* < 0.001) and a lack of an identity matrix (r = 0.022). Three factors were extracted from the 12 individual items (Table 5), explaining 46% of the overall variance. Only 11 items met the criteria (>0.40) for factor loadings; “I worry about trying a new sport or activity” did not load on any of the three factors. The first factor, *Perceived Confidence*, included items related to the student-athlete being confident in their body to complete activities and explained over 20% of the overall variance, with a reliability estimate of McDonald’s Omega α = 0.78 and inter-item correlations ranging from 0.36 to 0.63. The second factor, *Perceived Skill Ability*, explained an additional 16% of the variance, with a similar reliability estimate (McDonald’s Omega α = 0.72). These items focused more on specific skills necessary for participation in physical activity, with inter-item correlations from 0.08 to 0.46. The lowest correlation was between “I don’t really need to practice my skills, I’m naturally good” and “I understand the words that coaches and PE teachers use.” Besides the lowest correlation estimate, the remaining pairwise relationships were 0.30 to 0.46. As only two items were on the third factor, the corrected item total and inter-item correlation are the same (r = 0.53). McDonald’s Omega cannot be calculated when there are fewer than four items, so Cronbach’s α (0.69) was calculated. This factor, *Perceived Value*, consisted of two items regarding well-being, happiness, and their connection to being active. 

### 3.3. Physical Literacy Relative Ranking of Literacies

In the final subsection, the relative ranking of literacies, normality was violated, and principal axis factoring estimation was used. All remaining assumptions were met [Kaiser-Meyer-Olkin measure of sampling adequacy (*p* = 0.78), sphericity (Bartlett’s test χ^2^_36_ = 782.95, *p* < 0.001), and lack of an identity matrix (r = 0.004)]. Two factors were extracted from the nine individual items (Table 5), explaining nearly 68% of the overall variance. The first factor, *Academic Literacy and In-School Movement*, which included items related to reading, math, and activities within a school setting, explained over 28% of the overall variance, with a high level of reliability (McDonald’s Omega α = 0.85) and inter-item correlations ranging from 0.27 to 0.85. The second factor, *Outside of School Movement*, explained an additional 26% of the variance, with items specific to movement with friends and family. Again, as only two items were loaded onto this factor, Cronbach’s α (0.90) was used instead of McDonald’s Omega α, and the inter-item correlation was 0.82.

## 4. Discussion

This study aimed to assess the underlying psychometric properties of the PLAY*self* in a cohort of secondary school-aged athletes. The *PLAYself* is a self-reported measure of physical literacy with the subscales of environment, physical literacy self-description, relative ranking of literacies, and fitness. The hypothesis was supported by the fact that the scale is indeed multidimensional in nature. However, the factors did not align with similar work in other populations [18,20,21], which is expected during the validation process [38]. Jefferies et al. [20] published the first psychometric estimates of the PLAY*self* in 2021 in a sample of 8–14 year olds in Canada, showing unidimensionality in each of the three subsections (environment, self-description, and relative rankings of literacies). In other words, it was appropriate to find a summative score for each of the subsections, as the items measured similar constructs. Our results suggest that there are seven internal factors, not just the three subsections as previously published [20].

The first subscale of the PLAY*self* is environment. Rather than all six factors loading together, as demonstrated in previous work with younger children [20], this subscale broke into two distinctive components, consistent with work done in a young adult population [21]. The first factor contained water environments –water, ice, or snow. The second factor included the dry land environments of the gym, outside, and playground. This may be attributed to the fact that data were collected from a sample of students in the Southwest United States, many of whom live in a desert landscape. 

The second subscale of physical literacy self-description loaded into three separate factors. In recent work [21], in a sample of young adults (aged 18–25 years), physical literacy self-description loaded into only two factors. Both items, “I think being active is important for my health and well-being” and “I think being active makes me happier”, remained together on a single factor in both studies. In our work, a factor of perceived skill ability was retained to differentiate items of perceived overall confidence in activity from specific skill-based knowledge. The results differed the most in the final subsection of relative rankings. In Kleis et al. [21], three factors were extracted but not named. The current findings suggest that reading and math skills are important at school and home but not with friends, which is unsurprising. All reading- and math-related items loaded onto a single factor, including “movement at school”, which could be interpreted as after-school sports and aligns with this population. 

Both Jefferies et al. [20] and the current results suggest that the item “I worry about trying a new sport or activity” should be removed from the PLAY*self* self-description subsection. Jefferies et al. [20] hypothesized that the item may have a stronger link to a personality trait of being open to new experiences rather than the item’s original purpose of measuring motivation in children. This would be consistent with a secondary school-aged population. Given that these participants were engaging in an optional after-school sport, they are likely open or have been open to new experiences in the past, making this question less likely to discriminate between those that have high levels of motivation and those that do not. 

Our reliability results differ from those of Caldwell et al. [18]; however, that sample was with children aged 8–13. In the work by Caldwell et al. [18], reliability estimates (McDonald’s Omega α) were lowest in the PLAY*self* environment subsection (α = 0.65), while the highest internal consistency (α = 0.85) was seen within the PLAY*self* physical literacy self-description subsection. In the present work with student-athletes aged 13–18, PLAY*self* relative ranking of literacies factors (α = 0.85–0.90) had the highest internal reliability, and the lowest was within the PLAY*self* self-description (α = 0.69–0.78). This suggests younger children may have more variability in responses to access to different environments for activity [19]. In contrast, older adolescents may have more life experience, allowing them to participate in a variety of environments, as seen in the current work and that of Kleis et al. [21]. 

In the work by Eighmy et al. [19], with middle school students (6th to 8th grade) and parents, the lowest relative scores were in the environment section, which directly aligns with self-reported experiences of secondary school student-athletes. The lowest scores were seen in their perceived self-confidence in activities on ice and snow; nearly 25% of the sample had never tried physical activity in these environments. Again, this could be related to the geographical location of the study, which is a warm, southwestern state. 

While this work was conducted using rigorous methodologies on a novel population of secondary school student-athletes, it is not without limitations. First, it is possible that these participants did not fully understand the listed activities in the PLAY*self*. Second, the location of the study in a single Southwestern state and data collection in two cities across three secondary schools limited the generalizability of the findings. Also, while diverse secondary schools were recruited and student-athletes participated in this study, these student-athletes may not represent the overall study body characteristics and may be a more homogeneous cohort. There is no gold standard measure for physical literacy, and much of the existing work has focused on younger ages. The PLAY*self* presented many benefits, as it is self-reported and quick (about 5–10 min) for participants to complete. However, additional studies in larger and more diverse populations of secondary school-aged athletes and non-athletes should be conducted, including those athletes focusing solely on club sports or student artists participating in ballet or the circus.

Further, high levels of self-reported fitness may be explained by the fact that this was a sample of student-athletes actively participating in a school-sanctioned sport at the time of data collection. Other studies on US youth and young adults have not reported on the fitness question from the PLAY tools [26,39], but future studies should evaluate if this impacts different aspects of physical literacy. Additionally, to complete the validation process, a confirmatory factor analysis needs to be conducted with a new cohort of student-athletes in the secondary school set to ensure the factor structure is maintained. 

## 5. Conclusions

The PLAY*self*, a self-reported measure of physical literacy, demonstrated strong psychometric properties, including construct validity and reliability, in a cohort of secondary school student-athletes. Our results suggest that more than a single construct is measured in each subsection, and therefore, a summative score may not truly represent the complexities of physical literacy. More specifically, each factor should have its own score and be treated independently. 

## Figures and Tables

**Table 1 ijerph-21-01294-t001:** Participant Demographics and Sample Descriptive Statistics.

	Frequency (Percent)
Sex	
Female	75 (49.7)
Male	76 (50.3)
Race	
American Indian/Alaskan Native	7
Asian	5
Black	19
Hawaiian/Pacific Islander	3
White	111
Prefer Not to Say	3
Ethnicity	
Not of Hispanic, Latino/a/x, or Spanish origin	111 (77.1)
Mexican, Mexican American, Chicano/a/x	15 (10.4)
Another Hispanic, Latino/a/x, or Spanish origin	18 (12.5)
Puerto Rican	0
Cuban	0
Highest Completed Grade	
8th	34 (22.8)
9th	38 (25.5)
10th	40 (26.8)
11th	37 (24.8)
Type of Sport *	
Badminton	3
Baseball	2
Basketball	21
Cheerleading	10
Cross Country	5
Football	30
Golf	5
Hockey	0
Lacrosse	0
Mountain Biking	1
Soccer	11
Softball	11
Swim	1
Tennis	1
Track and Field	12
Volleyball	28
Wrestling	2
	*Mean ± Standard Deviation*
Age (years)	15.7 ± 1.2
Environmental Subsection	360.8 ± 118.0
Physical Literacy Self-Description Subsection	910.8 ± 125.5
Relative Ranking of Literacies Subsection	682.9 ± 150.4
PLAY*self* Score (sum of 3 subsections/27 items)	71.2 ± 10.6

* Percents not added as participants could select more than one option.

**Table 2 ijerph-21-01294-t002:** Physical Literacy Environment Subsection Frequencies and Percent of Responses.

	Frequency (Percent)				
	Never Tried	Not so Good	OK	Very Good	Excellent
Gym	1 (0.7)	2 (1.3)	22 (15.0)	74 (50.3)	48 (32.7)
Water	13 (8.8)	19 (12.9)	62 (42.2)	30 (20.4)	23 (15.6)
Ice	37 (24.8)	54 (35.8)	29 (26.2)	9 (6.0)	10 (6.7)
Snow	35 (23.5)	29 (19.5)	43 (28.5)	20 (13.4)	22 (14.8)
Outdoors	4 (2.8)	3 (2.1)	14 (9.7)	58 (40.0)	66 (45.5)
Playground	11 (7.4)	6 (4.1)	28 (18.9)	55 (37.2)	48 (32.4)

**Table 3 ijerph-21-01294-t003:** Physical Literacy Self-Description Subsection Frequencies and Percent of Responses.

	Frequency (Percent)			
	Not True at All	Not Usually True	True	Very True
It doesn’t take me long to learn new skills, sports or activities.	0	28 (22.2)	52 (41.3)	46 (36.5)
I think I have enough skills to participate in all the sports and activities I want.	0	23 (18.0)	53 (41.4)	52 (40.6)
I think being active is important for my health and well-being.	0	1 (0.7)	20 (13.9)	123 (81.5)
I think being active makes me happier.	0	1 (0.7)	21 (14.9)	119 (84.4)
I think I can take part in any sport/physical activity that I choose.	0	8 (6.2)	46 (35.7)	75 (58.1)
My body allows me to participate in any activity I choose.	0	11 (8.5)	41 (31.8)	77 (59.7)
I worry about trying a new sport or activity.	0	59 (41.5	77 (54.2)	6 (4.2)
I understand the words that coaches and PE teachers use.	0	5 (3.9)	49 (38.0)	75 (58.1)
I am confident when doing physical activities.	0	7 (5.3)	45 (33.8)	81 (60.9)
I can’t wait to try new activities or sports.	0	24 (18.5)	56 (43.1)	50 (38.5)
I’m usually the best in my class at doing an activity.	0	48 (36.4)	51 (38.6)	33 (25.0)
I don’t really need to practice my skills, I’m naturally good *.	31 (23.3)	64 (48.1)	29 (21.8)	9 (6.8)

* Presented as completed by responded (i.e., not recoded).

**Table 4 ijerph-21-01294-t004:** Physical Literacy Relative Ranking of Literacies Subsection Frequencies and Percent of Responses.

		Frequency (Percent)			
		Strongly Disagree	Disagree	Agree	Strongly Agree
Reading and writing are very important.	In school	4 (2.6)	2 (1.3)	55 (36.4)	90 (59.6)
	At home with family	5 (3.3)	22 (14.6)	77 (51.0)	47 (31.3)
	With friends	14 (9.3)	43 (28.5)	61 (40.4)	33 (21.9)
Math and numbers are very important.	In school	2 (1.3)	6 (4.0)	45 (30.2)	96 (64.4)
	At home with family	11 (7.3)	32 (21.3)	65 (43.3)	42 (28.0)
	With friends	17 (11.3)	49 (32.7)	55 (36.7)	29 (19.3)
Movement, activities, and sports are very important.	In school	3 (2.0)	5 (3.4)	38 (25.5)	103 (69.1)
	At home with family	0	6 (4.0)	50 (33.6)	93 (62.4)
	With friends	1 (0.7)	2 (1.3)	43 (28.9)	103 (69.1)

**Table 5 ijerph-21-01294-t005:** Item Loadings and Correlations for Each Factor of the PLAY*self* Scale.

Factor Name	PLAY*self* Item	Factor 1 Item Loadings	Factor 2 Item Loadings	Factor 3 Item Loadings	Item-Total Correlation	Scale Reliability
Environment Subsection						
Water Environment	In the water	0.64			0.54	0.83
	On the ice	0.88			0.74	
	On the snow	0.83			0.71	
Dry Land Environment	In the gym		0.41		0.39	0.71
	Outdoors		0.54		0.61	
	On the playground		0.44		0.52	
Self-Description Subsection						
Perceived Confidence	I think I have enough skills to participate in all the sport activities I want.	0.54			0.50	0.78
	I think I can take part in any sport/physical activity that I choose.	0.99			0.70	
	My body allows me to participate in any activity I choose.	0.61			0.61	
	I am confident when doing physical activities	0.45			0.53	
Perceived Skill Ability	It doesn’t take me long to learn new skills, sports or activities.		0.55		0.51	0.72
	I understand the words coaches and PE teachers use.		0.44		0.38	
	I can’t wait to try new activities or sports.		0.59		0.53	
	I’m usually the best in my class at doing an Activity.		0.59		0.61	
	I don’t really need to practice my skills, I’m naturally good.		−0.45		0.39	
Perceived Value	I think being active is important for my health and well-being.			0.64	0.53	0.70
	I think being active makes me happier.			0.59	0.53	
Relative Rankings of Literacies Subsection						
Academic Literacy and In-school Movement	Reading at school	0.55			0.50	0.85
	Reading at home with family	0.72			0.69	
	Reading with friends	0.63			0.63	
	Math at school	0.57			0.44	
	Math at home with family	0.72			0.74	
	Math with friends	0.71			0.73	
	Movement at school	0.64			0.42	
Outside of School Movement	Movement at home with family		0.51		0.82	0.90
	Movement with friends		0.60		0.82	

## Data Availability

Dataset available upon request from the authors.

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
