# Peer review of "Psychometric Properties of the PLAYself in a Cohort of Secondary School Student-Athletes"

_ijerph, 2024, doi:10.3390/ijerph21101294_

Round 1

Reviewer 1 Report

Comments and Suggestions for Authors

General comments: 
This manuscript examined psychometric properties of the PLAYself instrument in a sample of secondary school student-athletes.
Overall, this manuscript was well written which made it enjoyable and easy to read. With minimal revisions, I believe this manuscript is suitable for publication.

Specific comments: 
Introduction:

The introduction is well-written. It provides a nice overview of the topic of physical literacy/other background information, and it clearly identifies the gap in current literature.

Line 51-52: The researchers justify this population by highlighting the lack of literature. Maybe include an additional sentence about the importance for obtaining this information for secondary student-athletes specifically.

Methods: 
I think
the methods are very well written. The researchers did a nice job of writing in a concise manner yet providing enough details about how the student was performed. I believe I could reproduce this study with the information provided which cannot be said about many manuscripts these days!

Line 77-78: Can the authors clarify why they used ‘completed grade’ instead of the current grade of the student-athlete.

Lines 96-99: For readers not familiar with the PLAYself, it may be helpful to mention that the fitness item was not included because that’s how the original questionnaire is designed (as opposed to the researchers omitting it for various reseasons).

Line 97: I do not think the “)” needs to be there.

Results: 

Do the researchers have any information on the specific sports student-athletes were involved in? It may not be relevant but could be interesting to note. Additionally, did the researchers explore any differences in physical literacy by grade level (again, may not be relevant to the current study, but may be interesting to explore in the future). Otherwise, the results are nicely presented. The tables make it especially easy to visualize the data.

Discussion:

Again, the discussion is well written. The researchers do a great job comparing/contrasting their results with similar studies. However, I encourage the researchers to explore their results a little more in depth. For example, they state that “this is a very confident cohort”. Why do they think this is?; what role might athletics play in this?; do they think these results would be the same in another group of US secondary student-athletes?; etc.

Reviewer 2 Report

Comments and Suggestions for Authors

Dear authors.

First of all, I would like to thank you for the proposal you have prepared and sent, as it seems to me to be of great interest and relevance. Of course, it is necessary to praise the effort and work done.

In addition, I would like to make some contributions of little significance, but which can be easily dealt with and which are the following:

- 2.1. Study Participants and Study Design: Why did the students in the sample have to be involved in a sports season? I think this is something interesting that should be explained; with regard to the prior sending of the questionnaire, so that it could be reviewed by the families, I have the feeling that this could interfere with the results obtained, since it cannot be guaranteed that the families have not ‘worked’ with their children on the questionnaires. Why is this done? For what purpose?

- 2.2. Instrumentation: Why do the number of possible responses vary in the Likert scales? Would it not be more appropriate to try to have the same number of possible responses when dealing with the data as a whole (physical literacy); When talking about the non-inclusion of the score on the fitness level, this is repetitive (rows 96-99).

- 2.3. Data collection: I do not quite understand the reason why there are two data collection moments for students who are participating in sports from the same starting point. The longer time of practice could affect the results...

- 2.4. Data analysis: Why were the sections recoded with these scores, 600, 900 and 1200?

Best regards.

Reviewer 3 Report

Comments and Suggestions for Authors

The paper “Psychometric Properties of the PLAYSelf in a Cohort of Secondary School Student-Athletes” is directed to undertaking a psychometric eval of the play self tool in grade 8-12 student athletes, which fills a gap in the literature.

I read this paper with interest, and find it can make a good contirbutuion to the literature with revision. 

- missing some references for comparison 

- need to justify the athlete argument a bit more 

- need to justify an EFA when your language is CFA 

Abstract

The purpose was to extend validation to older age group, so the background is incomplete story for setting the rationale for the study. Need to add a line stating this rationale.

design of the PLAY Self tool was based upon the IPLA definition of physical literacy (Whitehead 2010)  not the 2015 abbreviated “public” definition of the Aspen Institute from Project Play. I see why you’d want to use the abbreviated def but it is not true to the origin of the construct, specifically  “reported measure of ability, confidence, and desire”. Please revise.  

The results and conclusions in abstract are nicely stated.

Introduction

The paper illustrating a relationship of PLAY Self  to PA, and that PL may be protective against PA erosion (in the pandemic) should be cited in the intro paragraph (ref- when the world stops- Houser et al). This not only substantiates the relationship of PL to elevated PA (and btw directly to the tool of interest), but PL may be protective  against PA erosion.

Line 29-30 implies a physically illiterate threshold – possible rephrase ?

The definition choice does not fit with the references cited, most of these referfences adopt the IPLA definition 33-34. Further the definition you cite arose from Aspen Project Play 2015 document “PHYSICAL LITERACY IN THE UNITED STATES A MODEL, STRATEGIC PLAN,  AND CALL TO ACTION”.  Please revise.  Further, the domains that you state are associated with the IPLA def, the Canadian Consensus statement (2015) as well as other consensus statements (Australia 2019, Ireland 2022 and England 2023) – so lines 33-39 should be coherent in this regard.

Lines 39-43 seem letter placed with the  first intro paragraph. And I would suggest adding the Danish work showing mediation effects of PL to PA to health, and PL directedly to health.

Line 46 – PLAY suite of tools was released in Banff, Canada at the first international physical literacy conference in 2013. And the tools were design for people aged 7 years and over, but original intent was for children and youth (to 18+), but clearly the PLAY SELF tool has one unidimensionally distinct set of items relating to literacies which doesn’t work well with people out of school.

Further there are over 25 studies that have reported on PL self description that you don’t cite here and its potential value. I don’t think you have to but your references are “biased” toward the physical activity and guideline achievement goals. Just a note to consider.

Line 48-51 – the PLAY self tool has been applied in older adults, and indeed adapted for older aged individuals esp by the Chinese. These should be mentioned to support your “gap” statement at line 51.

Line 50 – the need to associate with country should either be provided for each assessment or dropped.

Line 51 – you identify the “gap” in validation here to adolescent, high school student – athletes

-               

-              Line 53-54 “Adolescent student-athletes are in a stage of physical and emotional development,  making this a critical time”

o   I was not compelled by the statement above about the need for student -athlete validation, please explain why this is more important that additional validation at high school in students in general?  Are you implying for a performance excellence trajectory in sport contexts? As opposed to a general student’s trajectory to be active for life? (which is you foundation in the first two paragraphs). You could make a better  story about sport using the ADM and LTD3 documents and those pathways!

-               

-              Further, this “student-athlete” statement is not inclusive of others that are in performance development such as those that are “student artists” like those in development for circus or ballet. There is a paper on the special considerations for development of circus artists using PL by Stuckey et al that might be handy to ref here? Stuckey M, Richard V, Decker A, Aubertin P, Kriellaars D. Supporting Holistic Wellbeing for Performing Artists During the COVID-19 Pandemic and Recovery: Study Protocol. Front Psychol. 2021 Feb 4;12:577882. doi: 10.3389/fpsyg.2021.577882. PMID: 33613376; PMCID: PMC7889520.

o   High performance artists DO NOT identify as athletes by in large, FYI.

-              Line 54 – is it age appropriate you are seeking or athlete specific validation? Because the age range validated from the listed studies is 8-25! Which encompasses your ages but NOT specific to athletes.

-              Also if the athlete focus is indeed the focus of the study, there will have to be studies of non-student athletes, and professional athletes at this age as well.

Line 56  to 58. Well if you state this (alignment with existing domains), which I tend to agree with, then a EFA would be the wrong choice and a CFA should be deployed! Please defend.

61 – what was the purpose of the cross sectional larger study? The sentence in 61-63 is hard to decipher as to the intent of the larger study. Please explain.

Line 64 – what is a “rural city” ? versus and “urban city”  versus a large metropolitan area. I would revise this statement.

Line 65-67 If you are describing the student body of the one school using ref 19, how about the other schools? And also, does the average student body demographics represent the “student athlete “ group? There a plenty of studies indicating that the athletes at school of typically not representative of the student body as a whole!

Line 68 – so if the athletes were in Off season that would disregarded? Please explain. Also what if they were enrolled in a competitive sport that was not high school related? The were discarded? You state two semesters to be captured at line 101, so . why do you make the off season statement?

Line 73-74 – need consent document number and university listed.

Line 70 – minor assessment form? What is that, is this an assent form? So the participants provided written assent, and the parent provided written consent ? please be specific

Line 85  - there are many more constructs in the PL self description section that simply self-efficacy to PA! Revise.

Line 95-98 – most of these refences derived a sub-score for the three sub-scales (enviro, self-description and literacies) but not an overall score. Further, an overall score for the literacies section is suspect, unless it was limited to the movement questions. Please see the paper by Bremer et al “A cross-sectional study of Canadian children's valuation of literacies across social contexts”. If you look at this paper you could compare you findings to theirs!

Line 102 - Research team members met with coaches and student-102 athletes to describe the study and invite voluntary participation. So this survey wasn’t applied to the whole student body? So please list all the coaches you met with and what teams they represented.

Line 130-131 – which items used PFA and which used Max likelihood?

Line 121 – please defend EFA when you had entered with predefined construct domains.

Line 133 – please report the range of correlations among scales – you state “possible” so you need to defend the promax selection better than possible.

You state “ ordinal “ nature at line 136 – which I agree with . Both Mcdonalds and Cronback’s alpha are usually in alignment FYI. But use means and SD in the descriptives rather than median and IQR.  Please explain. I would pick one tradition (parametric or non-parametric) and stick with it.

Table 1- sex assigned at birth is what you mean here correct?

Table 1 – the designation of ethnicity is very odd “not of”

Table 1- I don’t understand highest completed grade when the age is narrow at 15.  But the grade completed would refer to lower and upper secondary. So to help me interpret the 11th grade completed student would be in grade 12?

It would have been handy to compare the “student athlete” to the student values, because if not different then that would be important to know. Does you larger data set have this?

Your table 2 – which I like to see this type of representation, now makes me realize that winter sports were clearly not represented in your school sports like you might get in the northern and mid western states. It would be handy to know what sports were recruited.

Table 3. The worry item is reverse coded. So do you reverse it for the table?

Your table 4 results need to be compared in discussion to the Bremer article, as they have very similar findings for general population students (>7000 measured in that study) and they show sex effects.

You probably need to show me the results of the comparing differences due to the sex of the participant. Especially since a sex and gender effect is known to exist! See Houser et al ACSM conference paper Girls just want to have fun! The competence-confidence-happiness cascade. North American Society for Pediatric Exercise Medicine (NASPEM). Saskatoon, SK. DOI:10.1123/pes.20220-0105 (peer reviewed published conference paper)

and the Bremer paper.

Table 5 - I don’t know how you derived the factor names – please explain how you aligned items to factor names.

Discussion

You will need a much more detailed comparison of your EFA results to the other psychometric papers,  when you make a statement “did not align” at line 271. Be specific. Also you need to tie back to the 4 domains of the IPLA def that you introduce! How did you know there was alignment?

An EFA versus a CFA versus a RASCH analysis are very different things. Confirmation of the alignment to the intended domains requires CFA!  Seven internal factors need to be aligned to the domains more clearly as they may simply be sub-domains of the major domains (and likely are). You should also peek at the recent model paper of Jen Agans (4d4d4all) which has a different domain organization.

Line 278-285 since your population was in a non snow and ice covered environment – desert as you call it, so the direct comparison to Jeffries is therefore not even suitable. Would it be preferred to state that the comparison is not appropriate and therefore not make the statements 278 - 281 and attribute it to age! Line 322 / 323 makes me realize that there is hockey in the southwest, especially Texas, and I wonder if you got any hockey players? And how they might differ.

The dataset should be added as a doi to this paper. That is the tradition of transparency emerging.

Line 350 informed consent and informed assent– you mean.
